# The Protective Effects of Ganoderic Acids from *Ganoderma lucidum* Fruiting Body on Alcoholic Liver Injury and Intestinal Microflora Disturbance in Mice with Excessive Alcohol Intake

**DOI:** 10.3390/foods11070949

**Published:** 2022-03-25

**Authors:** Ying-Jia Cao, Zi-Rui Huang, Shi-Ze You, Wei-Ling Guo, Fang Zhang, Bin Liu, Xu-Cong Lv, Zhan-Xi Lin, Peng-Hu Liu

**Affiliations:** 1National Engineering Research Center of JUNCAO Technology, Fujian Agriculture and Forestry University, Fuzhou 350002, China; 3190515002@fafu.edu.cn (Y.-J.C.); inaccelworld@gmail.com (Z.-R.H.); liubin618@hotmail.com (B.L.); lzxjuncao@163.com (Z.-X.L.); 2Institute of Food Science and Technology, College of Biological Science and Technology, Fuzhou University, Fuzhou 350108, China; 7200112062@stu.jiangnan.edu.cn (W.-L.G.); fangzh921@fzu.edu.cn (F.Z.); 3School of Clinical Medicine, Fujian Medical University, Fuzhou 350122, China; yousz2298@gmail.com; 4School of Food Science and Technology, Jiangnan University, Wuxi 214122, China

**Keywords:** *Ganoderma lucidum*, ganoderic acids, liver injury, intestinal flora, liver metabolism

## Abstract

This study aimed to investigate the protective effects of ganoderic acids (GA) from *Ganoderma lucidum* against liver injury and intestinal microbial disorder in mice with excessive alcohol intake. Results showed GA supplement significantly inhibited the abnormal elevation of the liver index, serum lipid parameters, aspartate aminotransferase and alanine aminotransferase in mice exposed to alcohol intake, and also significantly protected the excessive lipid accumulation and pathological changes. Alcohol-induced oxidative stress in the liver was significantly ameliorated by GA intervention through reducing the levels of maleic dialdehyde and lactate dehydrogenase and increasing the levels of glutathione, catalase, superoxide dismutase and alcohol dehydrogenase. Intestinal microbiota profiling demonstrated GA intervention modulated the composition of intestinal microflora by increasing the levels of *Lactobacillus*, *Faecalibaculum*, *Romboutsia*, *Bifidobacterium* and decreasing the *Helicobacter* level. Furthermore, liver metabolomic profiling suggested GA intervention had a remarkable regulatory effect on liver metabolism with excessive alcohol consumption. Moreover, GA intervention regulated mRNA levels of alcohol metabolism, fatty lipid metabolism, oxidative stress, bile acid biosynthesis and metabolism-related genes in the liver. Conclusively, these findings demonstrate GA intervention can significantly relieve alcoholic liver injury and it is hopeful to become a new functional food ingredient for the prevention of alcoholic liver injury.

## 1. Introduction

Alcoholic beverage is one of the most popular drinks consumed by billions of people worldwide. However, excessive alcohol consumption can disrupt liver metabolic function and homeostasis of oxidative stress, thereby adversely affecting human health, and has become a global health problem. Excessive alcohol consumption usually causes alcoholic liver damage (ALD), which may further develop into more severe hepatitis, liver fibrosis and even liver failure [1]. Alcohol metabolism is a complex process involving intestinal absorption, hepatic oxidative stress and lipid metabolism. Alcohol abuse may first result in excessive accumulation of acetaldehyde, a highly toxic metabolite that produces excessive active oxygen radicals and increases oxidative stress damage [2]. Besides, abnormalities in the metabolic pathway of alcohol dehydrogenase can generate a variety of harmful free radicals that attack lipids and cell membrane proteins, resulting in liver oxidative stress and lipid peroxidation [3]. Drug therapy is considered the current primary treatment for alcoholic liver injury, while most therapeutic drugs still have a series of unpleasant side effects. Therefore, looking for functional components with strong liver protective effects from natural food resources is a very promising strategy to decrease the risk of ALD.

*Ganoderma lucidum* (*Lingzhi*) is a well-known medicinal mushroom of “medicine and food homology” in China, and has been widely applied as a valuable medicine in Eastern countries for over 1000 years. The pharmacological activities of *G. lucidum* are derived from the various bioactive substances, including polysaccharides, proteins, polysaccharide–peptides, phenolic acids, triterpenoids, and ganoderma acids. Accumulating evidence has demonstrated that polysaccharides [4], triterpenoids [5] and ganoderma acids [6] from *G. lucidum* have good preventive effects on non-alcoholic fatty liver and hyperlipidemia. Ganoderic acids (GA) belonging to triterpenoids are the most important bioactive components in the *G. lucidum* fruiting body, and have been proved to have an extensive range of pharmacological features including antitumor, antioxidant, immunomodulatory, hypocholesterolemic, antimicrobial, anti-inflammatory, hypoglycemic and hypolipidemic properties [7,8]. It has been previously revealed that oral administration of *G. lucidum* ethanol extract (mainly consisting of ganoderic acids) ameliorated hyperlipidemia by regulating the expressions of key genes related to lipid and cholesterol metabolism. Moreover, it was also reported that oral administration of the aquatic extract of *G. lucidum* reduced the steatohepatitis induced by excessive drinking through anti-inflammation and antioxidant pathways, and reduce lipid accumulation in mouse livers [9]. However, despite the reported beneficial effects of ganoderic acids on the liver, the protective effects of ganoderic acids against the pathological process of ALD and its potential mechanism of action have not yet been fully elucidated.

Human health is strongly related to the homeostasis of gut microflora and the permeability of the intestinal barrier. Over the past few years, many studies have suggested that excessive drinking might cause the disorder of gut microbial homeostasis and the destruction of intestinal barrier permeability, resulting in the proliferation of harmful bacteria in the intestinal tract, the absorption of harmful metabolites, and the damage of liver metabolism function [10,11]. The “gut–liver–metabolite” axis plays an influential role in the pathological development of ALD. The intestinal microbiota of alcoholics are characterized by a lower proportion of *Faecalibacterium*, *Eisenbergiella* and *Akkermansia*, but a higher proportion of *Enterococcus*, *Escherichia-Shigella*, *Odoribacter*, *Parasutterella* and *Psychrobacter* [12,13,14]. A previous study has shown that *Lactobacillus*, *Bifidobacterium* and *Akkermansia* could restore intestinal microbial diversity and reduce the damage of oxidative stress in the intestine caused by alcohol metabolites such as acetaldehyde and free radicals [15]. Besides, excessive alcohol consumption also significantly changes the metabolites generated from intestinal microbiota, especially intestinal short-chain fatty acids (SCFAs) [16], whose level is highly linked with the structure of intestinal microbiota, the functions of liver metabolism. Hence, targeting the gut microbiota through diet would be the fit way to prevent or alleviate ALD. So far, few studies have attempted to characterize the beneficial effects of ganoderic acids on intestinal microbial composition and its relationship to the hepatoprotective effects in mice with excessive alcohol intake.

The current research was designed to investigate the protective effects of ganoderic acids intervention against ALD in mice with excessive alcohol intake, through high-throughput sequencing and liver metabolomics combined with metabolic pathway analysis. The connections between intestinal microbes and lipid metabolism relevant parameters were further revealed by correlation analysis and visualized by network, which could provide powerful theoretical proofs for exploiting functional foods to prevent ALD.

## 2. Materials and Methods

### 2.1. Raw Materials and Chemical Reagents

*G. lucidum* fruiting body was purchased from Fujian Agriculture and Forestry University, China. Total cholesterol (TC), triglyceride (TG), low-density lipoprotein cholesterol (LDL-C), high-density lipoprotein cholesterol (HDL-C), aspartate transaminase (AST), alanine transaminase (ALT), glutathione (GSH), catalase (CAT), maleic dialdehyde (MDA), superoxide dismutase (SOD), lactate dehydrogenase (LDH), alcohol dehydrogenase (ADH) and aldehyde dehydrogenase (ALDH) were purchased from Nanjing Jiancheng Bioengineering Institute (NJBI, Nanjing, Jiangsu, China). All other chemicals and reagents used in this experiment were of analytical reagent grade and purchased from Shanghai Sangon Biotech. Co., Ltd. (Shanghai, China).

### 2.2. Preparation of GA by Organic Solvent Extraction

*G. lucidum* fruiting body was dried by hot air and then pulverized, and the powder was obtained through a 60 mesh screen. The above powder was soaked in absolute ethanol with an ultrasonic extractor (solid–liquid ratio 1:20, power 100 W) at 70 °C for 90 min and then filtered. The ethanol extract of *G. lucidum* was concentrated under reduced pressure at 50 °C using a rotary evaporator, in order to remove the water and ethanol. Next, the ethanol extract was purified by organic solvent extraction by a variety of steps to obtain GA. Briefly, the concentration was dissolved in distilled water and extracted with equal volume of ethyl acetate. The organic solution supernatant (the ethyl acetate layer) was extracted with equal volume saturated sodium bicarbonate solution three times. The pH of sodium bicarbonate extract was adjusted to 2–3 with HCl (6.0 mol/L), and then extracted with equal volume ethyl acetate and washed with distilled water. The solution was then concentrated under pressure at 60 °C, and purified ganoderic acids were obtained by lyophilization for the animal experiments.

### 2.3. Compound Composition Analysis of GA by HPLC-QTOF MS

Compound composition analysis of GA was performed with high-performance liquid chromatography coupled with QTOF electrospray ionization MS. Chromatographic separations were performed with an Agilent 1260 high-performance liquid chromatography (Agilent, Santa Clara, CA, USA) on a ZORBAX SB-C18 column (4.6 mm × 250 mm, 5 μm; Agilent, Santa Clara, CA, USA). The mobile phase was a mixture of 0.01% acetic acid in water (A) and acetonitrile (B) using a gradient elution of 28%B–30%B at 0–25 min, 30%B–39%B at 25–50 min, 39%B–60%B at 50–60 min, 60%B–100%B at 60–61 min, 100%B–28%B at 61–65 min. Finally, the column was equilibrated at 28%B for 3 min. The column temperature was set at 30 °C, the flow rate was 1 mL/min, the wavelength was 252 nm, and the injection volume was 20 μL. The mass spectrometry detection was carried out in an Agilent 6530 QTOF electrospray ionization MS system (Agilent, Santa Clara, CA, USA) under negative ion electrospray mode. With regard to MS/MS analysis, the following operational parameters were used: drying N_2_ gas flow rate at 10 L/min and the temperature was maintained at 350 °C; nebulizer, 40 psig; capillary, 3500 V. An auto MS/MS was used to achieve the negative ion mode in *m*/*z* range of 60–800 at a scan rate of 1 spectrum/s, and fixed collision energies (5.00, 20.00, 40.00 eV). The mass spectrometer and HPLC system were controlled by Qualitative analysis 10.0 (Agilent, Santa Clara, CA, USA).

### 2.4. Animals and Experimental Design

Thirty-four specific pathogen-free (SPF) male Kunming mice (6-week-old) were purchased from Fujian Wushi Animal Co., Ltd. (Fuzhou, Fujian, China). All mice were kept in hygienic conditions (temperature of 24 ± 1 °C, humidity of 55 ± 5.0% and light from 08:00 to 20:00) and randomly had access to a normal chow diet and water. After 10 days of adaptation, all the experimental animals were stochastically separated into four groups for 6 weeks, including the control group (*n* = 8), model group (*n* = 10), GA-L group (12 mg kg^−1^ mg^−1^, *n* = 8) and GA-H group (36 mg kg^−1^ mg^−1^, *n* = 8). In this study, GA was re-suspended in distilled water and given orally at 10 a.m. every day as a curative treatment for 6 weeks. The control group and model group were also gavaged the same volume of saline as the GA group at the same time. The model group, GA-L group and GA-H group were gavaged with a dose of 50% ethanol (7.5 mL/kg) at 2 p.m. every day. Meanwhile, the mice in the control group were orally administered the same volume of saline. During the experiment, all mice were freely given sufficient feed and water and weighed weekly.

### 2.5. Sample Collection and Preparation

At the end of the experiment, the blood samples were collected from the heart of anesthetized mice after fasting for 12 h and then centrifuged at (3000× *g*, 10 min) to obtain the serum. The weights of the liver, kidney, and spleen were measured at once after the mice were sacrificed. Afterward, the cecal contents, the remainder of the liver and jejunum were quickly put in liquid nitrogen and kept at −80 °C until further analysis.

### 2.6. Biochemical Assays of the Serum and Liver Samples

Serum levels of TC, TG, HDL-C, LDL-C, AST and ALT were measured using commercial biochemical kits from NJBI (Nanjing, Jiangsu, China). The liver homogenate was prepared with normal saline by high-speed homogenizer on ice, and then the supernatant was obtained by centrifugation at 1000× *g* and 4 °C for 10 min. TC, TG, CAT, GSH, SOD, MDA, ADH and LDH levels of the liver were measured using commercial biochemical kits. All the kits were produced by NJBI (Nanjing, Jiangsu, China).

### 2.7. Pathological Analysis of Liver

The fresh liver samples were fixed in 4% paraformaldehyde solution overnight followed by dehydration through a series of ethanol solutions, embedded in paraffin and cut into 5 μm thick slices. The fabricated liver sections were stained by hematoxylin and eosin, and then scanned by an optical microscope (Olympus, Tokyo, Japan).

### 2.8. High-Throughput Sequencing of Intestinal Microbiome

Bacterial DNAs were extracted from cecal contents by Fast DNA SPIN Extraction Kit (Mobio Laboratories, Carlsbad, CA, USA) and quantified by NanoDrop Spectrophotometer (Thermo Fisher Scientific, Waltham, MA, USA). The V3-V4 hypervariable regions of the 16S rRNA gene were amplified by a forward primer 341F and a reverse primer 805R, and then sequenced based on Illumina MiSeq platform at Shanghai Majorbio Co., Ltd. (Shanghai, China) according to our previous research [6].

The sequencing data were screened and clustered into operation taxon units (OTUs) with a 97% identity threshold by QIIME 2.0 software. Compared with the sequences in the GreenGenes database (Ver. 13.8; LBNL, SF, USA), the microbial phylotypes were identified at the genus level, and the relative abundance of each genus was obtained. Principal component analysis (PCA) and clustering analysis were performed using SIMCA-14.1 software (Umetrics, Umea, Sweden) at the genus level. Variation analyses of different experimental groups were conducted using linear discriminant analysis coupled with effect size (LEfSe) (threshold >2) based on Galaxy Online Platform (http://huttenhowe.sph.harvard.edu/galaxy/, accessed on 13 December 2021). The associations of the key intestinal microbes with biochemical or metabolic parameters were revealed by Spearman’s rank correlation coefficient through R software (Ver. 3.3.3, AKL, New Zealand) and visualized as a network through Cytoscape software (Ver. 3.6.0, Bethesda, MD, USA).

### 2.9. UPLC-QTOF MS Analysis of Liver Metabolome

Liver metabolites were extracted with mixed organic solvents (acetonitrile:methanol:water = 2:2:1) and shaken with a vortex shaker for 1 min. After centrifugation at (10,000× *g*) for 15 min at 4 °C, the supernatant was transferred to a centrifuge tube and evaporated to dryness at 37 °C under nitrogen flow. Dried samples were redissolved in 200 μL of 50% acetonitrile and centrifuged at 4 °C (12,000× *g*, 10 min). The supernatant was injected into the UPLC-QTOF/MS system (Agilent, Santa Clara, CA, USA) for the analysis of liver metabolome. Liver metabolites were separated by Agilent 1290 Infinity UPLC system (Agilent, Santa Clara, CA, USA) equipped with a BEH amide column (2.1 mm × 100 mm, 1.7 μm, Waters) and identified through 6530 QTOF electrospray ionization mass spectrometer (Agilent, Santa Clara, CA, USA).

The raw MS/MS data obtained from the UPLC-QTOF MS system were analyzed by Mass Profiler Professional software (Agilent, Santa Clara, CA, USA). The peak intensity was output for multivariate statistical analysis through MetaboAnalyst 5.0 (http://www.metaboanalyst.ca/, accessed on 13 December 2021), including PCA, PLS-DA and OPLS-DA. The VIP values and *p* values determined by two-tail *t*-tests (VIP > 1.0 and *p* < 0.05) were used as filtering criteria for the selection of potential biomarkers. Then through the Human Metabolome Database (http://hmdb.ca/, accessed on 13 December 2021), the exact quality and typical MS/MS fragments were compared to identify biomarkers. Pathway analysis of liver biomarkers with significant differences between the model and GA-H groups was performed at the online platform MetaboAnalyst 5.0.

### 2.10. Quantitative RT-PCR

Total hepatic RNA was extracted by a commercial RNA extraction kit (RNAiso Plus, Code No. 9108) provided by Takara Biomedical Technology (Beijing, China) Co., Ltd., and then reverse-transcribed into cDNA using a commercial cDNA kit with gDNA Eraser (Takara, Beijing, China). qPCR was completed in StepOne Real-Time quantitative PCR System (Applied Biosystems, Foster City, CA, USA) with SYBE Green Ex Taq™ II q-PCR kit (Takara, Beijing, China). The PCR was conducted as follows: initial activation 95 °C for 30 s, denaturation 95 °C for 5 s, annealing 55 °C for 30 s, extension 72 °C for 30 s, 40 cycles. The mRNA expressions level was normalized to the 18S rRNA gene. In this study, the 2^−ΔΔCt^ method was used to analyze the relative expression levels of related genes. The qPCR primers used in this study were listed in Table 1.

### 2.11. Statistical Analysis

All data were expressed as the mean ± standard deviation. The significances of results were analyzed by one-way analysis of variance (ANOVA) with GraphPad Prism 7.0. Values were expressed as mean ± SD, and ^##^ *p* < 0.01 and ^#^ *p* < 0.05, versus the model group; ** *p* < 0.01 and * *p* < 0.05, versus the control group.

## 3. Results

### 3.1. HPLC-QTOF/MS Analysis of the Compound Composition of GA

The compound composition of GA was analyzed through HPLC-QTOF/MS under the negative ionization mode because triterpenoids and ganoderic acids have one or more hydroxyl and/or carboxylic acid groups, which have higher sensitivity and selectivity in negative ionization mode. The total ion chromatogram at negative ion mode showed that GA mainly contains 23 compounds (the top 23 according to peak area percentage) (Appendix A), which were then tentatively identified by retention times, pseudomolecular ions and their ion-fragmentation patterns observed in the MS/MS spectrum. Information on the accurate masses of the pseudomolecular ions of the peaks and their fragmentation patterns from reference standards and previous literature were also used to identify the 23 compounds in the GA [17,18,19]. Retention time and mass spectral data as well as peak assignments for compounds identified in negative ionization mode are shown in Table 2. A total of twenty-three triterpenoids were finally identified in GA, among which ganoderic acid C2 (peak 5), ganoderic acid G (peak 6), ganoderenic acid B (peak9), ganoderic acid Xi (peak 10), ganoderic acid B (peak 11), ganoderic acid K (peak 14), ganoderic acid A (peak 15), ganoderic acid H (peak 16), ganoderic acid E (peak 19), 12-hydroxyganoderic acid D (peak 20), ganoderic acid D (peak 21), ganoderic acid F (peak 22) and 12-acetoxyganoderic acid F (peak 23) were the representatives with high peak area percentage.

### 3.2. Effects of GA Intervention on the Body Growth Performance

As shown in Figure 1, at the beginning of the experiment, there was no obvious difference in the body weights of mice among different experimental groups. However, after oral administration of 50% alcohol (*v*/*v*) for 6 weeks, the body weights of mice in the model group were obvious lighter than those of the control group (*p* < 0.05), indicating that excessive alcohol intake might be harmful to the body’s metabolic function. GA intervention significantly reversed the abnormal reduction of body weight induced by excessive alcohol intake in a dose-dependent manner. The liver plays a crucial role in regulating alcohol metabolism because more than 80% of alcohol is consumed in the liver. The liver index is one of the most sensitive parameters of alcohol metabolism and can be a valid reflection of the degree of liver damage to some extent. In this study, the liver weight and liver index, kidney index and spleen index of the mice in the model group were significantly higher than those of the control group (*p* < 0.05), suggesting that excessive alcohol intake for 6 weeks may cause liver, kidney and spleen injury in mice. Compared with mice of the model group, oral administration of GA at 12 and 36 mg/kg b.w. significantly reduced the organ index of mice with excessive alcohol intake (*p* < 0.05), suggesting GA is effective in preventing alcohol-induced damage to the liver, kidneys and spleen.

### 3.3. Effects of GA Intervention on Serum Biochemical Parameters

Apparent increases in serum TC, TG and LDL-C (*p <* 0.01) levels were observed after 6 weeks of excessive alcohol consumption compared with mice of the control group, nevertheless the serum HDL-C level was dramatically decreased in mice with excessive alcohol intake (*p <* 0.01) (Figure 2), indicating that the liver metabolic function was obviously destroyed in the mice that were drinking too much alcohol. After six weeks of GA supplementation, the abnormal serum levels of TC, TG and LDL-C were strongly reduced compared with mice in the model group (*p* < 0.05), while the serum HDL-C level was prominently increased, especially in mice orally given high-dose GA (36 mg/kg b.w.). Moreover, as the two most important and sensitive biomarkers of liver metabolic function, serum ALT and AST levels were markedly increased in mice with alcohol intake (*p* < 0.05), reflecting the existence of liver inflammation or liver injury. Notably, oral supplementation of low-dose and high-dose GA (12 mg/kg b.w. and 36 mg/kg b.w.) significantly down-regulated alcohol-induced abnormal increase in serum ALT and AST levels, suggesting that alcohol-induced liver injury can be attenuated by daily GA supplementation in a dose-dependent manner.

### 3.4. Effects of GA on Liver Biochemical Parameters and Histopathological Features

Compared with the mice without alcohol intake, the mice with excessive drinking (the model group) had abnormal higher levels of hepatic TC, TG, MDA, LDH, but lower levels of hepatic GSH, CAT, SOD, ADH (*p <* 0.01) (Figure 3A). Similar to the control group, GA intervention reversed the abnormal changes in hepatic lipid accumulation and liver injury caused by alcohol exposure (*p <* 0.05). Histopathological analysis of liver sections displayed that the hepatocytes of mice in the control group had evident liver lobule, orderly arrangements of liver cell cords, round central nucleus, distinct cytoplasm and cell borders (Figure 3B).

In contrast, the model group mice showed the disordered hepatic cord, blurred boundary of hepatocytes, and slight necrosis of the cells around the hepatocyte central vein. It was also observed that some hepatocytes were steatosis and inflammatory cells infiltrated in the model group. After GA intervention for 6 weeks, the hepatic cords were arranged radially around the central vein, the necrosis of liver cells alleviated significantly, and the structure of hepatic has lobules become clear, indicating that the treatment of GA can markedly improve the degree of alcoholic liver lesion.

### 3.5. Effects of GA Intervention on the Composition of Intestinal Microbiota

Principal component analysis (PCA) and hierarchical clustering analysis (HCA) were used to analyze the intestinal microbial compositions among different experimental groups (the control, model and GA-H groups) (Figure 4A). The result of PCA illustrated the model group was clearly distinguished from the control group (Figure 5A), suggesting the composition of intestinal flora was profoundly affected by excessive alcohol intake. Specifically, there was a noteworthy structural shift of intestinal flora along the positive direction of PC2 in the model group. On the contrary, the supplement of high-dose GA changed the shift of intestinal flora in the negative direction of PC2 induced by alcohol but made the intestinal microbiota move in the negative direction of the PC1. In addition, the hierarchical clustering dendrogram illustrated that the composition of the intestinal microbiota of the GA-H group was apparently other than that of the model and control groups (Figure 4B). These results indicated that a high dose of GA treatment could modulate the intestinal microbiota composition of mice with excessive alcohol intake.

To appraise the effects of high-dose GA intervention on the components of gut microbiota at the genus level, the differences of the relative abundance of gut microbial phylotypes between the control and model groups, the model and GA-H groups, were performed by LEfSe analysis (LDA score > 2). As shown in Figure 5A,B, the relative proportions of 19 key intestinal microbial genera were significantly changed in mice with excessive alcohol intake, including 5 apparently increased microbial genera and 14 markedly decreased microbial genera. The mice with excessive alcohol intake have a higher proportion of *Enterococcus*, *Escherichia-Shigella*, *UCG-007*, *Defluviitaleaceae*_UCG-011, and *norank_o_Oscillospiraceae* than those without alcohol intake (the control group). Compared with the model group, the mice without alcohol intake (the control group) were characterized by higher amounts of *norank_f_Muribaculaceae*, *Alloprevotella*, *Bacteroides*, *Pseudomonas,* and *Butyricicoccus*, etc. On the other hand, there were 17 microbial phylotypes at the genus level with discriminative features between the model and GA-H groups (Figure 5C,D). Notably, high-dose GA intervention increased the relative abundance of *Lactobacillus*, *Faecalibaculum*, *Romboutsia*, *Bifidobacterium*, *Clostridium_sensu_stricto_1*, UCG-010, NK4A214_group, *Christensenellaceae_R-7_group*, *Turicibacter*, etc., in mice with excessive alcohol intake. By comparison, the relative abundance of *Helicobacter* was significantly decreased by GA intervention.

### 3.6. Correlations of the Key Microbial Phylotypes with the Biochemical Parameters

The heatmap and network based on Spearman’s correlation analysis showed the association between the key microbes with the biochemical parameters (Figure 6). It is noteworthy that *Helicobacter* (significantly enriched in mice with excessive alcohol intake) exhibited a positive relation to the serum TG level and a correlation with serum HDL-C level. Interestingly, the genera of *NK4A214*_group and *Clostridium*_sensu_stricto_1, whose levels were enriched in the GA-H group, were negatively related with the serum AST level and hepatic TG, MDA levels, of which *Clostridium*_sensu_stricto_1 was also negatively associated with the serum TG and LDL-C levels. Besides, the genera of *Lactobacillus*, *Bifidobacterium*, *Romboutsia*, *Turicibacter*, and *Faecalibaculum* were correlated negatively with the lipid metabolic parameters. Notably, *Bifidobacterium* showed a positive association with liver SOD level and norank_f_UCG_010 a negative association with serum TG, LDL-C, AST and hepatic MDA levels.

### 3.7. Effects of GA Intervention on Liver Metabolome in Mice with Excessive Alcohol Intake

In the current research, UPLC-QTOF-MS was performed to detect the difference in liver metabolomics and revealed the protective mechanism of GA intervention against excessive alcohol-induced liver injuries. To visualize the difference among different experimental groups, multivariate statistical analysis was performed to compare the metabolomic data of liver samples. Score plots of PCA and PLS-DA demonstrated that the metabolic spectrum of difference was separated at both ESI+ (Figure 7A,B) and ESI− (Figure 8A,B) ions modes, indicating that supplementary with high-dose of GA reversed the metabolite changes caused by excessive alcohol consumption. Furthermore, OPLS-DA score plots and (Figure 7C and Figure 8C) demonstrated there was obvious separation between the model and GA-H groups. The S-plots of OPLS-DA exhibited the differences in liver metabonomic profile between the model and GA-H groups (Figure 7D and Figure 8D). Moreover, there were a total of 180 metabolites (ESI+ mode: 104 metabolites & ESI− mode: 76 metabolites) between the GA-H and model group which were screened with a threshold of VIP > 1.0 and *p* > 0.05. In detail, the heatmap showed 35 liver metabolites (betaine [M118T344_6], guanidinosuccinic acid [M176T389], (−)-riboflavin [M377T302_2], diazepam [M285T221], (+)-aphidicolin [M285T57], 7-keto-8-aminopelargonic acid [M109T51_2], etc.) were increased and 69 liver metabolites (glycerophosphocholine [M258T396_3], trehalose [M360T402], cellobiose [M325T401_2], melezitose [M543T428], DL-proline [M116T362_4], 8-hydroxyquinoline-2-carbonitrile [M153T223_2], 1h-1,2,4-triazol-3-amine [M85T401_2], etc.) were decreased in the positive-ion mode. Additionally, 32 liver metabolites (including 2,5-furandicarboxylic acid [M111T215_2], D-proline [M114T364_1], allantoin [M157T295_2], tauroursodeoxycholic acid [M498T216], 3-hydroxybutyric acid [M103T316], xanthosine [M283T308], etc.) were increased and 44 liver metabolites (including L-(+)-lactic acid [M89T314_5], uridine [M243T281], lactulose [M161T400], palatinose [M221T401], sucrose [M341T397], 9-fluorenone [M179T362], etc.) were decreased in the negative ion mode compared with the model group (Figure 7E and Figure 8E).

The effects of high-dose GA intervention on the metabolic pathway of excessive alcohol intake were analyzed in terms of enrichment of metabolic pathways of differential liver metabolites (Figure 7F and Figure 8F). In the ESI+ and ESI− mode, metabolic pathways enrichment manifested that starch and sucrose metabolism, cysteine and methionine metabolism, arginine and proline metabolism, glycine, serine and threonine metabolism, glycolysis/gluconeogenesis, pyruvate metabolism, tryptophan metabolism, tyrosine metabolism, taurine and hypotaurine metabolism, riboflavin metabolism, pentose and glucuronate interconversions, ascorbate and aldarate metabolism, galactose metabolism, pyrimidine metabolism, fructose and mannose metabolism, and purine metabolism were the main metabolic pathways changed by high-dose GA treatment compared with the model group.

### 3.8. Effects of GA Intervention on Liver mRNA Levels in Mice with Excessive Alcohol Intake

To elaborate the potential mechanism of action by which GA intervention protects against alcoholic liver injury, mRNA expression levels of related genes to alcohol metabolism-related enzymes, fatty lipid metabolism, oxidative stress and bile acid biosynthesis were analyzed by RT-qPCR. Compared with mice of the control group, alcohol intake significantly inhibited the mRNA levels of aldehyde dehydrogenase 2 (*ALDH2*), heme oxygenase-1 (*HO-1*) and NADPH quinineoxidoreductase-1 (*NQO-1*) (*p <* 0.05), but significantly decreased the mRNA levels of nuclear factor erythroid-2-related factor 2 (*Nrf2*), catalase (*CAT*), superoxide dismutase-1 (*SOD-1*), glutathione peroxidase (*GSH-Px*), low-density lipoprotein receptor (*Ldlr*), apolipoprotein E (*ApoE*), cholesterol 7α-monooxygenase (*Cyp7a1*), carnitine palmitoyl transferase-1 (*CPT-1*), acyl-CoA oxidase 1 (*Acox1*), long-chain-fatty-acid-CoA ligase 1 (*Acsl1*) (*p <* 0.01) (Figure 9). Conversely, high-dose GA intervention significantly upregulated these expression levels (*p <* 0.05). The liver mRNA levels of hydroxymethylglutaryl-CoA reductase (*Hmgcr*) and CCAAT/enhancer binding protein alpha (*C*/*EBP-α*) were higher than those in mice of the control group. High-dose GA intervention distinctly decreased the mRNA levels of *Hmgcr* and *C/EBP-α* caused by excessive alcohol intake. Although there were no apparent differences in the liver mRNA levels of alcohol dehydrogenase 2 (*ADH2*) and bile salt export pump (*Bsep*) between the model group and the control group, high-dose GA intervention clearly increased the mRNA expressions of *ADH2* and *Bsep* in mice with excessive alcohol intake (*p <* 0.05).

## 4. Discussion

Alcoholic liver injury, a global public health problem, is threatening human wellness which is mainly caused by excessive alcohol consumption. Alcohol metabolism damage can be divided into two parts, one is direct damage resulting from the toxic effect of excessive acetaldehyde on liver cells, the other is indirect damage caused by other external factors. Alcohol could directly damage the liver which results in the development of ALD. Furthermore, the liver is more susceptible to oxidative stress, accumulation of ROS, lipid peroxidation, and other foreign factors, which trigger a more complex pathology that eventually leads to liver injury [20]. To replicate the liver function injury induced by excessive alcohol consumption, mice were given oral intragastric administration, which is one of the common models of ALD. This approach not only overcomes the aversion most rodents have to alcohol, but also imitates oral pathways, ingestion, and subsequent metabolic processes in humans [21]. It is well-known that *G. lucidum* possesses various excellent beneficial effects on liver function and lipid metabolism. The goal of this study was to investigate the protective effects of ganoderic acids on ALD and intestinal microbial disorder in mice with excessive alcohol intake. In particular, ganoderic acids inhibit liver oxidative stress, regulate lipid metabolism and improve intestinal microbial disorder, thus alleviating ALD in mice. The results of this study show that ganoderic acids, a potential active substance, can mitigate liver injury and intestinal microbial disorder caused by excessive alcohol intake.

In the current research, the body weight growth of mice was inhibited by excessive alcohol consumption and affected appetite, digestion, and absorption. Compared with the mice in the control group, the liver and kidney indexes of mice with ALD were significantly elevated. Furthermore, the increase of liver index in the model group was also reflected in the liver weight increase and body weight reduction, indicating the possible hepatic steatosis or hepatic hypertrophy [22]. Besides, the increase in kidney weight was attributed to the accumulation of alcohol metabolites excreted by kidney oxidation [23]. By contrast, excessive alcohol intake not only inhibited the growth of the spleen and the body weight but also reduced the spleen index, which was consistent with a previous study [24]. In short, oral administration of GA increased the lower body weight, spleen index, and improved the abnormal elevated liver and kidney indexes. In addition, GA supplementation not only significantly reduced serum lipid parameters (TC, TG and LDL-C) and increased HDL-C levels, but also effectively reduced liver lipid parameters (TC and TG) and the liver MDA level, suggesting that GA intervention could improve alcohol-induced liver injury by reducing lipid peroxide. As an important parameter reflecting the potential antioxidant capacity of the body, the content of MDA can better reflect the degree of liver peroxidation lesion. When liver cell damage is caused by excessive drinking, the level of lactate dehydrogenase in the liver is elevated and the permeability of the cell membrane increases, resulting in the enhancement of serum AST and ALT levels [25]. The results discovered that AST and ALT concentrations in the model group were much higher than other groups, which were used as an indicator of liver injury. Under normal conditions, the body has a complete antioxidant defense system, and the oxygen free radicals produced by the body are in dynamic balance with the antioxidant defense system [26]. In case of long-term and excessive drinking, antioxidants and antioxidant enzymes are consumed in large numbers to remove excess free radicals produced by the body, which damages the antioxidant defense system, ultimately resulting in functional damage to the liver. In this study, compared with the model group, the liver *SOD*, *GSH-Px*, and *CAT* activities were visibly elevated in GA-treated mice, implying that GA had a strong antioxidant ability which maintained the redox balance disrupted by alcohol exposure.

The interaction between the liver immune system and microbiota is limited in healthy individuals. However, ALD could induce intestinal bacteria overgrowth and intestinal permeability increases. Harmful microorganisms and their metabolites increase and more easily penetrate the intestinal barrier, then transfer to the liver, promoting liver injury [27]. For example, *Helicobacter*, a potentially virulent pathogen, was rich in the model group but reduced its quantity in the GA-H group in this study. It could disrupt the intestinal barrier and intestinal microbes, promoting the development of intestinal inflammation [28]. In addition, the data show that GA-H treatment greatly increases the relative abundance of *Lactobacillus*, *Caldicoprobacter*, and *Bifidobacterium*. Of which, *Lactobacillus*, a type of probiotic, has many beneficial effects on many diseases, including attenuating multi-organ inflammation [29], reversing insulin resistance and hepatic steatosis [30,31]. Furthermore, *Bifidobacterium* and *norank_f_Flavobacteriaceae* are the main bacterial genus in intestinal microorganisms which are involved in and enhance bile acid metabolism. It has been proved that *Bifidobacterium* promotes the conversion of primary bile acids into secondary bile acids [32,33]. Besides, *Lactobacillus*, *Caldicoprobacter* are the main genus associated with lactic acid production and reducing pH in the gut, which promotes the production of short-chain fatty acids (SCFAs) by bacteria and maintains intestinal health [34,35]. The result was consistent with the previous report that ethanol has been shown to increase fecal pH and disrupt gut microbiota homeostasis [34]. Moreover, *Lactobacillus* and *Bifidobacterium* encode *ADH* and *ALDH* to mediate the metabolism of ethanol and acetaldehyde, respectively, reducing the accumulation of acetaldehyde in the intestinal tract, which can improve the intestinal barrier function, showing a protective action on the development of alcohol-induced liver injury in mice [36]. On the other hand, oral administration of a high dose of GA markedly increased the relative proportion of producers of SCFAs in mice with excessive drinking, which perform the functions of mucoprotection, immune hemostasis and metabolism, including *Faecalibaculum*, *Bifidobacterium*, *unclassified_c_Clostridia*, *Ruminiclostridium*, *unclassified_o_Coriobacteriales* and *Christensenellaceae_R-7_group*, which are positively correlated with acetic acid and increase total SCFAs production [37,38]. It was reported that *Unclassified_c_Clostridia* has the ability to increase propionic acid production [39]. In addition, *Christensenellaceae_R-7_group*, a well-known producer of butyric acid, plays a crucial anti-inflammatory role in the gut [40]. Interestingly, *Ruminiclostridium* was also negatively associated with the abundance of lipopolysaccharide biosynthesis [41], implying that it may play a part in reducing intestinal inflammation. The relative abundances of *Clostridium_sensu_stricto_1* and *Oscillospirales* have been found to decrease with the presence of liver injury recently [42,43], and it was consistent with the result of this study that they were increased after the high-dose of GA intervention for 6 weeks. The previous research showed the relative abundance of genus *Romboutsia*, which was clearly increased in the GA-H group, was positively correlated to the body weight and HDL-C and negatively related to TC, TG, LDL-C and ALT [44]. Moreover, *Turicibacter* is positively correlated with tryptophan, indole acid, which may regulate tryptophan metabolism [45]. The present study showed that alcoholism would lead to impaired tryptophan metabolism [46]. Therefore, we speculate that GA may change the composition of the gut microbiome, thereby reducing the risk of liver injury and lipid metabolism disorders.

The liver is the most crucial organ for biosynthesis and biotransformation in the body. Liver metabolomics based on UPLC-QTOF/MS was used to elucidate the mechanism of action of GA intervention on liver injury induced by excessive alcohol consumption. High-dose GA intervention distinctly regulated the concentrations of metabolites. Compared with the model group, the liver metabolites of D-fructose, cellobiose, lactulose, palatinose, trehalose and melezitose which take part in starch and sucrose metabolism, were strongly reduced by high-dose of GA supplement, which may counteract sucrose-induced steatosis [47]. Taurodeoxycholic acid (TUDCA), a derivative of ursodeoxycholic acid (UDCA), also has similar functions in liver disease. Furthermore, recent studies have suggested that UDCA/TUDCA can stimulate the secretion of alkaline bile by inducing the post-transcriptional effect of transporters such as *Bsep* and *MRP2* [48]. Allantoin, a diureide of glyoxylic acid, was significantly increased in the liver with GA intervention, compared with mice of the model group. It was reported that allantoin could improve diabetic symptoms, inhibit oxidative damages and promote mitochondrial biogenesis of the liver in HFD/STZ-induced diabetic mice [49], which was consistent with the result of this study. Moreover, GA intervention effectively upregulated N-acetylcysteinamide, pyruvate and betaine levels in the liver. In detail, N-acetylcysteinamide and pyruvate are effective scavengers of ROS [50,51] and betaine can improve oxidative defense by significantly increasing the expression of *SOD*, *CAT, Nrf2* and *HO-1* [52,53]. Furthermore, betaine can efficiently improve the body weight gain caused by HFD and decrease the content of TC, TG, and malondialdehyde [45]. It was indicated that high-dose GA intervention may enhance the N-acetylcysteinamide and pyruvate generation to alleviate ROS and lipid metabolism in the liver.

Furthermore, a higher level of (−)-riboflavin in the liver of the GA-H group mice was detected and identified. The water-soluble vitamin riboflavin participates in many key metabolic reactions, making it important for normal cell function and growth [54]. Additionally, riboflavin could deficiency inhibit fatty acid oxidation in mitochondria in the liver [55]. Therefore, high-dose GA intervention could induce sufficient riboflavin in the liver to obtain liver health. It was also reported that both borrelidin and 3-hydroxybenzaldehyde have a vascular protective effect, which showed a higher level in the liver of GA-H group mice [56,57]. In addition, the contents of 3-hydroxybutyric acid and D-proline, xanthosine with antitumor effects were also significantly upregulated in the GA-H group [58,59]. Besides, a high dose of GA also elevated the level of allyl isothiocyanate, a potential therapeutic agent for obesity and insulin resistance, not only improves lipid accumulation and inflammation in a fatty liver via *Sirt1/AMPK* and *NF-κB* signaling pathways, but also induces *Nrf2* activation and increases *HO-1* expression as a liver protective agent [60]. Therefore, the mRNA expressions of *Nrf2* and *HO-1* expression should be further investigated.

By contrast, alcohol is transported to the liver for reduction, where the alcohol and lactic acid compete with each other for reductase, which causes lactic acid to accumulate in the liver [61]. Hepatic L-(+)-lactic acid level was significantly reduced under the administration of a high dose of GA in the current study. Besides, the indolelactic acid and DL-lactate accumulate in large quantities in mice with alcohol-induced liver disease, were also decreased by GA-H intervention [62,63]. High-dose GA intervention also reduced the levels of glyceraldehyde and 3′,5′-cyclic inosine monophosphate in the liver, of which glyceraldehyde-derived advanced glycation end products (AGEs) trigger vascular inflammation and accelerate the development of diabetic atherosclerosis, and 3′,5′-cyclic inosine monophosphate butyrate could cause an increase in stress in phenylephrine contracted mouse aortic rings [64,65]. It was indicated that high-dose GA intervention could decrease the potential biomarker to ameliorate the ALD. On the one hand, metabolites 9-fluorenone and sulfasalazine, which increase liver ROS levels, were dramatically upregulated in the model group [66,67]. On the other hand, excessive drinking significantly upregulated the metabolites 1,2,4-benzenetriol, 1,2,3-benzenetriol, and doxorubicin related to oxidative stress, which damaged the body’s antioxidant defense system (including reducing the activity of *GSH-Px* and *SOD* and increasing the levels of AST, ALT and MDA) [68,69,70]. The current research demonstrated GA-H may improve the ROS levels through modulating the concentrations of the above metabolites. Interestingly, N-acetyl-d-lactosamine is an important marker of various cancers [70]. Glycerophosphocholine was found to be accompanied by Alzheimer’s disease in the brain [71], and its metabolites could modulate atherosclerosis and thus the risk for cardiovascular disease [72,73]. They all showed lower levels in the liver of the GA-H group mice. It was demonstrated that the GA intervention could effectively deplete the potential harmful metabolites in the liver to ameliorate the ALD in this study.

To elucidate the protective mechanism of GA intervention on liver injury induced by excessive alcohol consumption, the RT-qPCR was conducted to detect the mRNA levels of liver genes that were related to oxidative stress and lipid metabolism. Generally, about 90% of alcohol metabolism occurs in the liver. First, alcohol is converted to acetaldehyde by *ADH* and Cytochrome P450 2E1 (*CYP2E1*). The acetaldehyde is then oxidized to acetic acid by *ALDH2*. Eventually, the acetic acid is metabolized to carbon dioxide and water through the citric acid cycle [74]. The mRNA level of *ALDH2* was significantly reduced in the model group, indicating that excessive acetaldehyde accumulated in the hepatocytes. However, GA intervention apparently ameliorated the reduction trend caused by excessive alcohol intake. On the one hand, the transformation of oxidized nicotinamide adenine dinucleotide (NAD+) to reduced NADH leads to a decrease in NAD+/NADH ratio, which is beneficial to the accumulation of hepatic TG and fatty acid synthesis. On the other hand, acetaldehyde induces oxidative stress, increases ROS, reduces the content of *GSH-Px* and inhibits the activity of *CAT* and *SOD* [75]. *Nrf2-Keap-ARE* pathway is an important endogenous oxidative stress pathway. Under normal physiological conditions, *Nrf2* binds to *Keap1* and inactivates it. However, under oxidative stress, *Nrf2* uncoupled with *Keap1*, transferred to the nucleus, and bound to antioxidant response elements (ARE), inducing the expression of a series of antioxidant (including *SOD*, *CAT* and *GSH-Px*) and cell-protective proteins (including *HO-1* and *NQO-1*) genes [76]. *Nrf2/Keap1/ARE* regulates *GSH* levels by upregulating *GSH* synthetic and regenerative enzymes. *SOD* can effectively eliminate the active oxygen free radicals in the body, meanwhile, *SOD* and *CAT* have a synergistic effect to remove H_2_O_2_. *NQO-1* prevents ROS generation through the redox cycle. *HO-1* is a downstream gene regulated by *Nrf2*, a defense protein, which protects cells from oxidative stress by catalyzing heme degradation. Compared with the model group, GA supplement obviously upregulated the mRNA levels of *Nrf2* and downstream *ARE* (*HO-1* and *NQO1*), as well as *SOD*, *GSH-Px* and *CAT*-related genes expression [77]. Thus, we preliminarily speculated that GA intervention could improve ALD by regulating hepatic oxidative stress.

To further understand the effects of over-dose alcohol consumption on liver injury, the mRNA levels related to cholesterol (*LDLr*, *ApoE*, *Cyp7a1*, *Hmgcr*, *Bsep*) were measured. Cholesterol can be obtained from the diet or synthesized intracellularly by the HMG-CoA reductase pathway. The liver maintains cholesterol homeostasis through transportation of cholesterol, synthesis and excretion of cholesterol and bile acids (BAs). LDL-C combines with *Ldlr* to carry the cholesterol esters (CE) into cells through endocytosis, which participates in the synthesis of BAs. *Ldlr* can also prevent the interaction of *ALDH2* macrophages with AMPK, which leads to impaired cholesterol hydrolysis and lipid deposition in lysosomes [78]. *ApoE* can form lipoproteins with lipids other than LDL, including cholesterol, TG and phospholipids, which make lipoprotein enter cells for metabolic decomposition. *ApoE* plays an important role in cholesterol and TG transport and metabolism. *ApoE* deficiency is prone to increase the levels of TC and TG [79]. *Hmgcr* is a crucial rate-limiting enzyme that controls the synthesis rate of cholesterol in the liver, and *Cyp7a1*, a pivotal synthase gene, participated in the first step of primary bile acid biosynthesis [6]. The upregulation of *Cyp7a1* gene would promote the conversion of cholesterol in BAs and increase the excretion of BAs [80]. Alcoholic feeding reduced the expression of BAs synthesis genes but promoted BAs deposition, increased hepatic TC and TG, and liver injury. *Bsep*, a bile salt transport, mediates the transport of BAs in vitro [81]. The reduction of *Ldlr*, *ApoE*, *Cyp7a1*, *Hmgcr* and *Bsep* levels in liver mRNA resulting in alcoholism was improved by supplementation of high-dose GA. Therefore, GA supplementation ameliorated the metabolic disorder from the transportation, synthesis and excretion of cholesterol and BAs.

The liver is an important organ that regulates the absorption, synthesis and oxidation of fatty acids. Excessive alcohol consumption disturbs lipid metabolism through the above pathways, which promotes the accumulation of fatty acids in the liver and the development of ALD. High-dose GA supplementary markedly increased liver mRNA levels of lipid metabolism-related genes (*Acox1*, *CPT-1* and *Acsl1*) and decreased liver mRNA levels of *C/EBP-α*. Oral GA intervention significantly reduced liver TC accumulation and inhibited adipocyte differentiation and adipogenesis by inhibiting *C/EBP-α* expression. In addition, *C/EBP-α* expression showed a positive relationship with the serum level of HDL-C [82]. As previously reported, *Acsl1* affects the biosynthesis and β-oxidation of fatty acids, which was decreased by alcohol intake [47]. Besides, *CPT-1* is a key rate-limiting enzyme in fatty acid β-oxidation, which is consistent with this study [83]. Therefore, GA treatment increased fatty acid oxidation by promoting the expression of *Acsl1* and *CPT-1*. Moreover, GA can also promote the expression of *Acox1* and increase fat metabolism, thus reducing TC levels [84]. In short, GA intervention can reduce hepatic fatty acids by regulating the synthesis, oxidation and metabolism of fatty acids in the liver.

## 5. Conclusions

This study aimed to investigate the protective effects of ganoderic acids (GA, the main triterpenoids in the *G. lucidum* fruiting body) against ALD and intestinal microbial disorder induced by excessive alcohol intake. Oral administration of GA obviously prevented alcoholic liver injury partly through ameliorating oxidative stress and lipid metabolism abnormality in mice with excessive alcohol intake. Besides, GA showed the potential to ameliorate intestinal microbial disturbance in mice with excessive alcohol consumption. The possible protective mechanism of GA intervention in the course of ALD was also explored by liver metabolomics and RT-qPCR. This study preliminarily suggests that dietary supplementation with GA could ameliorate ALD possibly by modulating intestinal microbiota and liver metabolic function and regulating the expressions of key genes related to alcohol metabolism, fatty lipid metabolism, oxidative stress, bile acid biosynthesis and metabolism. This study reveals that GA has potential profitable effects in taking precautions against alcohol-induced liver injury, and is estimated to be a promising functional food ingredient. In further studies, we need to illustrate the protective mechanism of GA intervention on ALD through multiomics technology.

## Figures and Tables

**Figure 1 foods-11-00949-f001:**
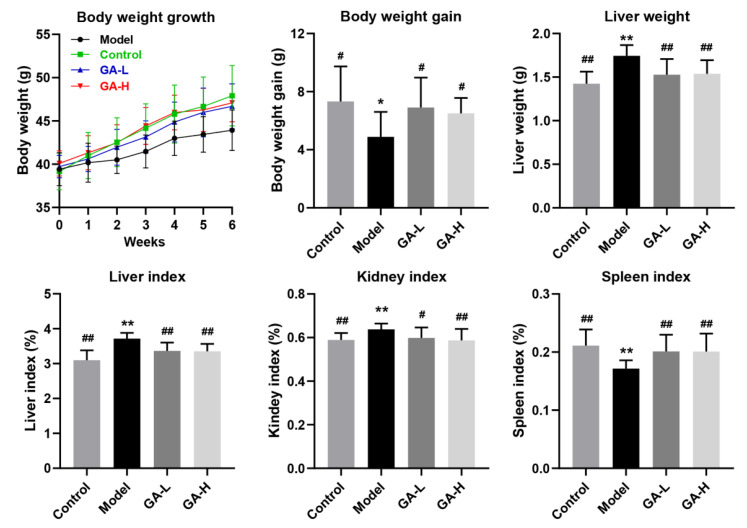
Effects of GA intervention on the body weight, liver weight, organ index (liver, kidney and spleen index) in mice with excessive alcohol intake. Values were expressed as mean ± SD, and ^##^
*p* < 0.01 and ^#^
*p* < 0.05, versus the model group; ** *p* < 0.01 and * *p* < 0.05, versus the control group.

**Figure 2 foods-11-00949-f002:**
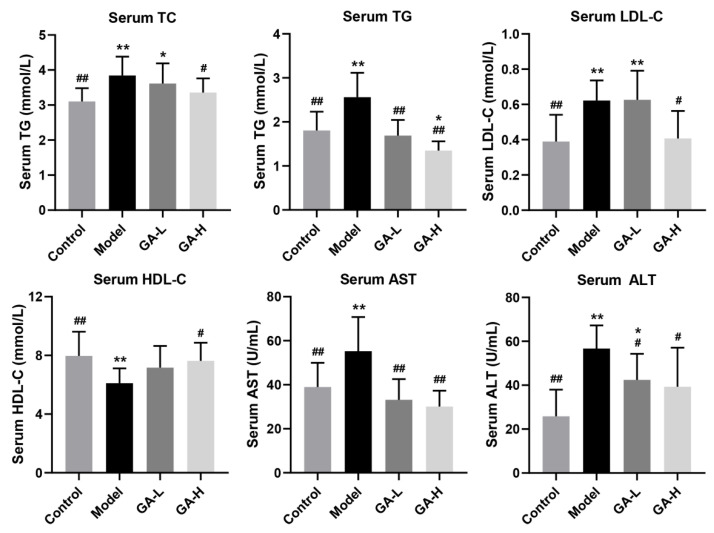
Effects of GA intervention on the serum biochemical parameters (TC, TG, LDL-C, HDL-C, AST and ALT) in mice with excessive alcohol intake. Values were expressed as mean ± SD, and ^##^
*p* < 0.01 and ^#^
*p* < 0.05, versus the model group; ** *p* < 0.01 and * *p* < 0.05, versus the control group.

**Figure 3 foods-11-00949-f003:**
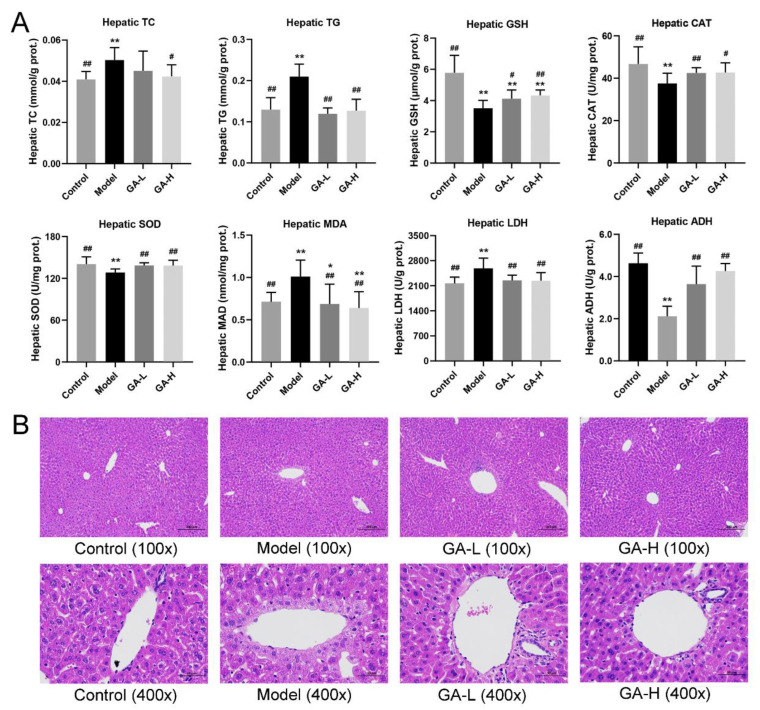
Effects of GA intervention on the liver biochemical parameters (TC, TG, GSH, CAT, SOD, MDA, LDH and ADH) (**A**) and liver histopathological features (**B**) in mice with excessive alcohol consumption. Values were expressed as mean ± SD, and ^##^
*p* < 0.01 and ^#^
*p* < 0.05, versus the model group; ** *p* < 0.01 and * *p* < 0.05, versus the control group.

**Figure 4 foods-11-00949-f004:**
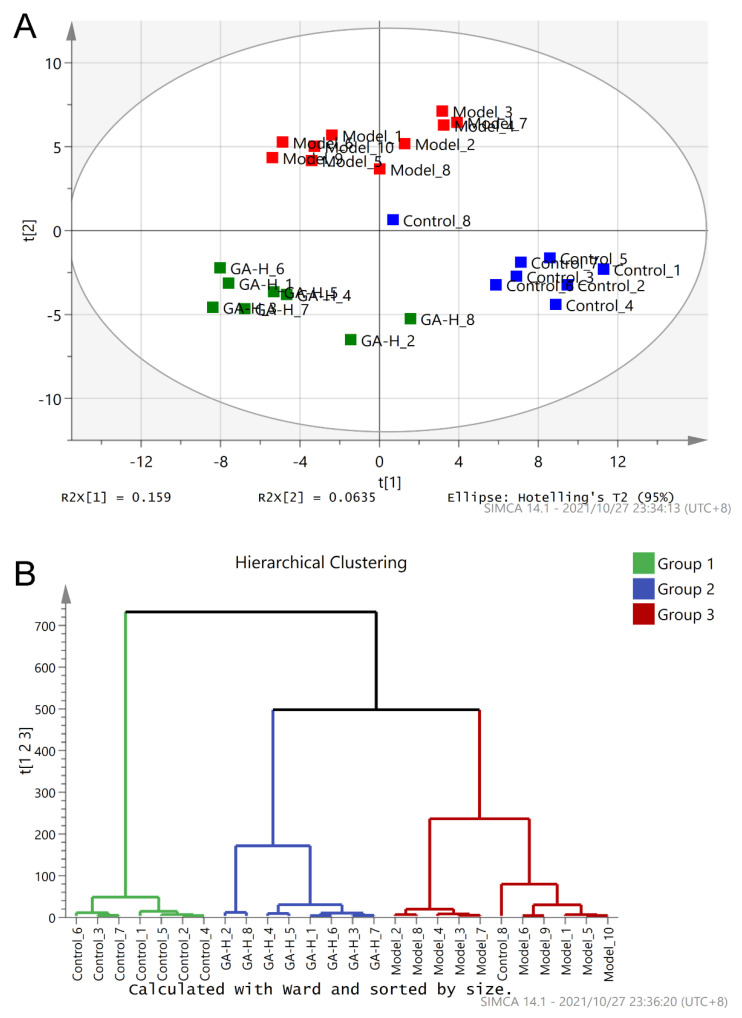
Regulation of GA intervention on the intestinal microflora in mice with excessive alcohol consumption for six weeks. (**A**) Principal component analysis (PCA) score plot; (**B**) hierarchical clustering analysis of intestinal microflora of different experimental groups.

**Figure 5 foods-11-00949-f005:**
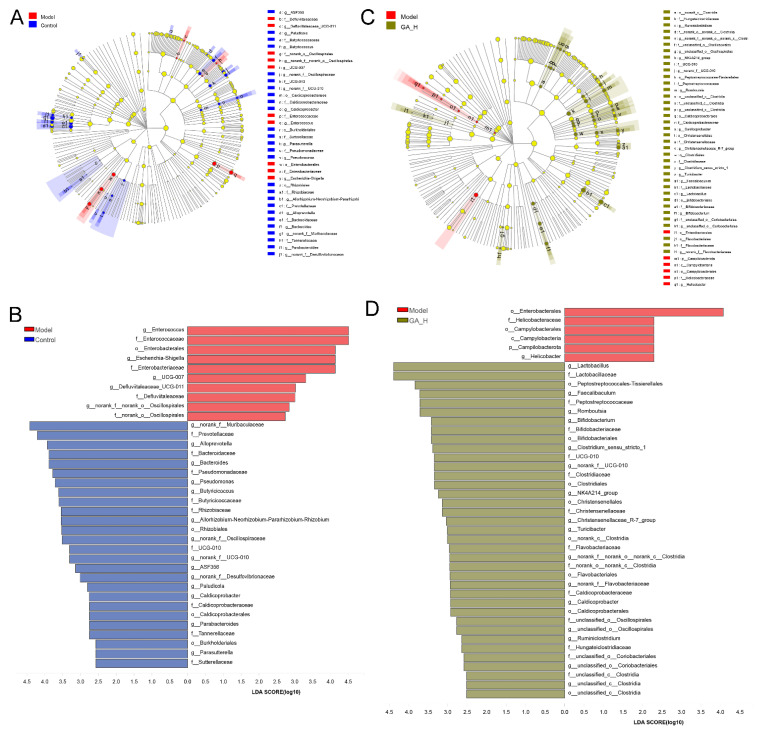
Characterization of microbiomes in mice of the control, model and GA-H groups by the linear discriminant analysis (LDA) effect size (LEfSe) method. (**A**,**B**) Taxonomic representation of statistically and biologically consistent differences in mice of the control, model and GA-H groups. (**C**,**D**) Histogram of the LDA scores (log10) calculated for features with differential abundance in mice of the control, model and GA-H groups.

**Figure 6 foods-11-00949-f006:**
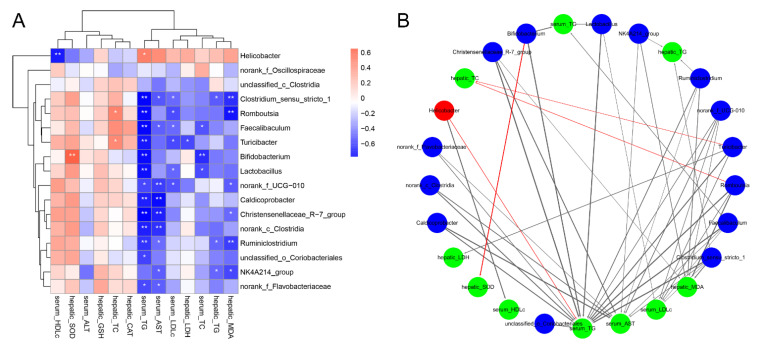
Correlation of intestinal microbes with biochemical parameters. (**A**) Heatmap of correlation between the biochemical parameters and the key microbial phylotypes at the genus level. (**B**) Correlation network between the biochemical parameters and the key microbial phylotypes. Red nodes: the key intestinal microbial phylotypes reduced by high-dose GA intervention; blue nodes: the key intestinal microbial phylotypes increased by high-dose GA intervention; green nodes: the biochemical parameters. The red lines represent positive correlations, and the black lines represent negative correlations. Line widths represent the strength of correlation. Only the significant edges were drawn in the network (|r| > 0.6, FDR adjusted * *p* < 0.05, ** *p* < 0.01).

**Figure 7 foods-11-00949-f007:**
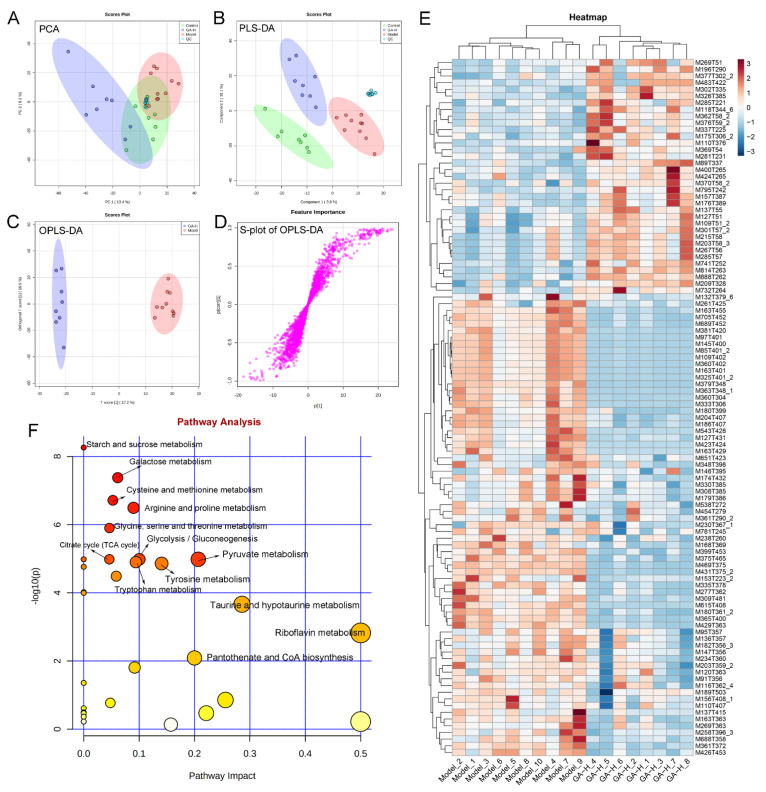
Effects of high-dose GA administration on liver metabolome in mice with excessive alcohol intake revealed by UPLC-QTOF/MS-based metabonomics in ESI+ mode. (**A**) PCA score plot; (**B**) PLS-DA score plot; (**C**) OPLS-DA score plot; (**D**) S-loading plot based on OPLS-DA analysis; (**E**) heatmap of liver metabolites with significant differences between groups (VIP value > 1.0, *p* < 0.05) between the model and GA-H groups; (**F**) metabolic pathway prediction based on the KEGG database.

**Figure 8 foods-11-00949-f008:**
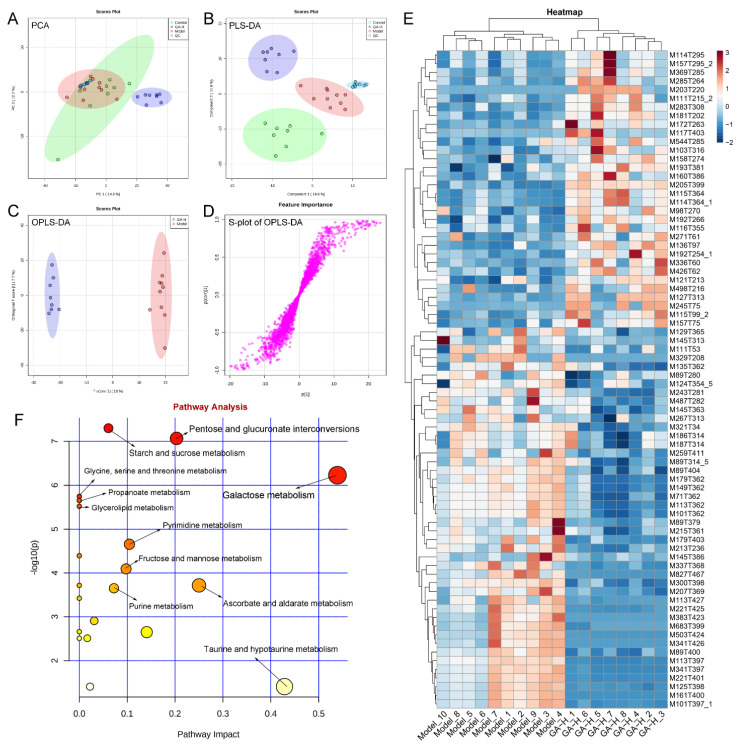
Effects of high-dose GA administration on liver metabolome in mice with excessive alcohol intake revealed by UPLC-QTOF/MS-based metabonomics in ESI− mode. (**A**) PCA score plot; (**B**) PLS-DA score plot; (**C**) OPLS-DA score plot; (**D**) S-loading plot based on OPLS-DA analysis; (**E**) heatmap of liver metabolites with significant differences between groups (VIP value > 1.0, *p* < 0.05) between the model and GA-H groups; (**F**) metabolic pathway prediction based on the KEGG database.

**Figure 9 foods-11-00949-f009:**
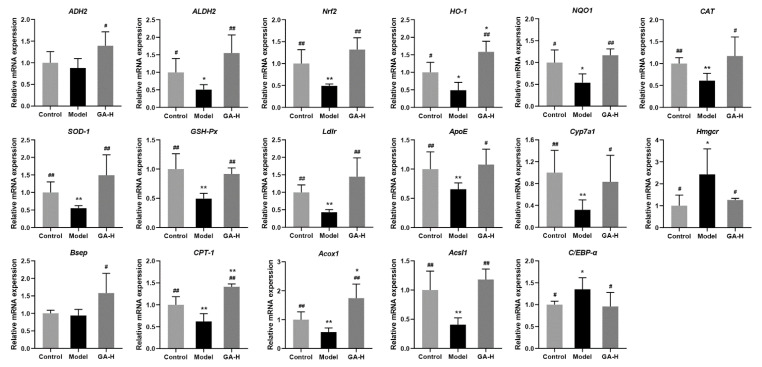
Effects of high-dose GA administration on the mRNA levels of oxidative stress and lipid metabolism correlative genes in livers of mice with excessive alcohol consumption. Values were expressed as mean ± SD, and ^##^ *p* < 0.01 and ^#^ *p* < 0.05, versus the model group; ** *p* < 0.01 and * *p* < 0.05, versus the control group.

**Table 1 foods-11-00949-t001:** Primer sequences for quantitative real-time PCR analysis.

Gene	Forward Primer (5′−3′)	Reverse Primer (5′−3′)
*Ldl* *r*	ATGCTGGAGATAGAGTGGAGTT	CCGCCAAGATCAAGAAAG
*Cyp7a1*	CCTTGGGACGTTTTCCTGCT	GCGCTCTTTGATTTAGGAAG
*Acox1*	GCCTGCTGTGTGGGTATGTCATT	GTCATGGGCGGGTGCAT
*ADH2*	AACGGTGAGAAGTTCCCAAAA	ACGACCCCCAGCCTAATACA
*ALDH2*	ATCCTCGGCTACATCAAATCG	GTCTTTTACGTCCCCGAACAC
*SOD-1*	TTG GCC GTA CAA TGG TGG	CGC AAT CCC AAT CAC TCC AC
*GSH-Px*	GGGACCCTGAGACTTAGAGC	AATCCGTACTAGCGCTCACA
*CAT*	TCA CCC ACG ATA TCA CCA GA	AGC TGA GCC TGA CTC TCC
*Nrf2*	GGGACCCTGAGACTTAGAGC	AATCCGTACTAGCGCTCACA
*HO-1*	AACAAGCAAKCCCAGTCTATGC	AGGTAGCGGGTATATGCGTGGGCC
*NQO1*	AACAAGCAAKCCCAGTCTATGC	AGGTAGCGGGTATATGCGTGGGCC
*Acl1*	CACTTCTTGCCTCGTTCCAC	GTCGTCCCGCTCTATGACAC
*CPT-1*	TCCATGCATACCAAAGTGGA	TGGTAGGAGAGCAGCACCTT
*ApoE*	AKCCGCTTCTGGGATTACCT	TCAGTGCCGTCAGTTCTTGTG
*C/EBP-α*	GAACAGCAACGAGTACCGGGTA	GCCATGGCCTTGACCAAGGAG
*Hmgcr*	TGCTGGTGCTATCAAAGG	GCAGATGGGATGACTCGA
*Bsep*	TCTGACTCAGTGATTCTTCGCA	CCCATAAACATCAGCCAGTTGT
*18S*	AGTCCCTGCCCTTTGTACACA	CGATCCCAGGGCCTCACTA

**Table 2 foods-11-00949-t002:** Characteristics of twenty-three peaks from the GA identified by HPLC-MS/MS in the negative ionization mode.

No	Rt (min)	UVλ_max_ (nm)	Formula	Assigned Identity	Precursor ion [M-H]^−^	Fragment Ions	Reference
1	7.89	257	C_30_H_46_O_8_	Ganoderic acid L	533.3101	515.2899, 497.2899, 405.2756, 129.0531, 87.0479	[17]
2	8.56	255	C_30_H_46_O_8_	12-hydroxyganoderic acid C2	533.3024	515.2899, 497.2874, 485.2865, 467.2719, 453.2908, 423.2751, 405.2756, 303.1559	[18]
3	10.39	256	C_30_H_40_O_8_	Elfvingic acid A	527.2595	509.2453, 465.2562, 421.2668, 317.1699	[18]
4	14.09	255	C_30_H_44_O_8_	Ganoderic acid η	531.2866	513.2777, 129.0527, 111.0425	[17]
5	19.29	256	C_30_H_46_O_7_	Ganoderic acid C2	517.3029	499.2965, 455.3075, 437.2979, 302.1813, 287.1616, 195.1028	[17]
6	20.95	254	C_30_H_44_O_8_	Ganoderic acid G	531.2929	513.2783, 469.2913, 451.2769, 436.2610, 319.1892, 265.1389, 249.1467	[18]
7	22.14	-	C_30_H_38_O_8_	Ganosporeric acid A	525.2464	507.2344, 451.2106, 129.0529, 495.1996, 229.1176	[19]
8	24.11	256	C_30_H_42_O_8_	Ganodoeric acid C6	529.2715	511.2592, 493.2432, 481.2115, 467.2702, 449.2608, 437.2232, 317.1724, 303.1524	[18]
9	27.05	249	C_30_H_42_O_7_	Ganoderenic acid B	513.2810	451.2839, 436.2592, 287.1642, 249.1462	[18]
10	27.99	-	C_30_H_42_O_7_	Ganoderic acid Xi	513.2764	495.2726, 465.2214, 451.2825, 383.2162, 331.1900, 235.1690, 151.1109, 73.0285	[19]
11	29.98	253	C_30_H_44_O_7_	Ganoderic acid B	515.2975	497.2840, 453.2940, 438.2717, 420.2620, 303.1926, 263.1626, 249.1457, 195.1362	[18]
12	31.98	263	C_30_H_42_O_7_	Ganoderic acid AM1	513.2782	495.2653, 451.2793, 436,2567, 421.2316, 249.1460	[18]
13	33.11	254	C_32_H_46_O_9_	Ganoderenic acid K	571.2791	553.2698, 538.2423, 511.2628, 467.2710, 303.1897, 265.1393	[18]
14	34.48	256	C_32_H_46_O_9_	Ganoderic acid K	573.3030	555.2904, 511.2988, 469.2920, 451.2807, 302.1843, 265.1405	[18]
15	37.86	253	C_30_H_44_O_7_	Ganoderic acid A	515.2923	497.2807, 453.2919, 435.2819, 195.0978	[17]
16	38.65	254	C_32_H_44_O_9_	Ganoderic acid H	571.2862	553.2768, 511.2671, 467.2768, 437.2306, 423.2668, 303.1578	[18]
17	41.61	250	C_27_H_36_O_6_	Lucidenic acid F	455.2348	425.1831, 395.2175, 383.2162, 301.1748, 247.1287, 149.0581	[18]
18	42.14	-	-	12-hydroxy-3,7,11,15,23-pentaoxo-lanost-8-en-26-oic acid	527.2552	509.2541, 465.2643, 435.2168, 301.1433	[18]
19	43.23	254	C_30_H_40_O_7_	Ganoderic acid E	511.2671	493.2484, 449.2638, 434.2381, 285.1442	[17]
20	45.26	256	C_30_H_42_O_8_	12-hydroxyganoderic acid D	529.2702	511.2629, 493.2510, 449.2618, 434.2406, 301.1764	[18]
21	47.54	256	C_30_H_42_O_7_	Ganoderic acid D	513.2782	495.2657, 451.2789, 301.1766, 283.1649, 247.1302, 193.1199	[18]
22	50.92	260	C_32_H_42_O_9_	Ganoderic acid F	511.2602	493.2537, 449.2638, 434.2404, 247.1307, 149.0509	[18]
23	54.92	250	C_32_H_42_O_9_	12-acetoxyganoderic acid F	569.2656	511.2525, 509.2514, 479.2054, 465.2617, 435.2144	[17]

## Data Availability

Not applicable.

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
