# Peer review of "The Protective Effects of Ganoderic Acids from Ganoderma lucidum Fruiting Body on Alcoholic Liver Injury and Intestinal Microflora Disturbance in Mice with Excessive Alcohol Intake"

_foods, 2022, doi:10.3390/foods11070949_

Round 1
Reviewer 1 Report
The article is interesting. However, I have few comment listed below.
Lines 129, 141, 135, 188, 205 and 207: For all the mentioned devices or softwares, apart from the name and model, all manufacturer's data should be given, i.e. manufacturer's name, city, state - e.g. in the case of the USA and country. There are no cities on the mentioned lines.
Lines 177, 178, 183, 193 and 194: A similar note as above. Complete manufacturer data is missing. In some cases, even the name of the manufacturer, e.g. of the software.
Line 138: In the sentence beginning "A auto MS/MS was .." should be "An auto MS/MS was ...".
Line 134: Please state the injection volume and the wavelength in nm at which the GA was detected and quantified.
Chapter 2.3: Was the calibration curve made for the chromatographic determination of GA content? Please include it in the supplementary material.
Line 144: It should be "were kept" and not "were keep".
Chapter 2.4: Please provide the approval number of your local ethics committee to conduct animal testing.
Lines 158 and 166: Please enter the centrifugation parameters in "g" unit.
Line 200: Please delete "at" before brackets.
Lines 253 and 254: Please write the names of the compounds in the middle of the sentence with a lowercase letter (12-hydroxyganoderic acid D, 12-acetoxyganoderic acid F).
There is something wrong with the page numbering. In the center of the document, the pages are counted from the beginning again.
Author Response
Response to Editor & Reviewers
Journal: Foods
Manuscript ID: foods-1622379
Original title: The protective effects of ganoderic acids from Ganoderma lucidum fruiting body on alcoholic liver injury and intestinal microbial disorder in mice with excessive alcohol intake
Revised title: The protective effects of ganoderic acids from Ganoderma lucidum fruiting body on alcoholic liver injury and intestinal microflora disturbance in mice with excessive alcohol intake
Dear editors and reviewers,
On behalf of my co-authors, I thank you very much for giving us an opportunity to revise the manuscript (Manuscript ID: foods-1622379). We appreciate your positive and constructive comments and suggestions on our manuscript. All of the comments are valuable and helpful for improving our manuscript, as well as the important guiding significance to our future research. Therefore, we have discussed all the comments carefully and made some corrections to meet with approval. The revised content has been marked in red or underlined in the revised manuscript. The main corrections in the manuscript and the response to the editor and reviewers’ comments are as follows:
Editor and Reviewers' comments (Q) and our responses to the comments (A):
Reviewer 1: The article is interesting. However, I have few comment listed below.
Q1: Lines 129, 141, 135, 188, 205 and 207: For all the mentioned devices or softwares, apart from the name and model, all manufacturer's data should be given, i.e. manufacturer's name, city, state - e.g. in the case of the USA and country. There are no cities on the mentioned lines.
A1: Thanks for the reviewer’s comment. We have revised the manuscript accordingly.
Q2: Lines 177, 178, 183, 193 and 194: A similar note as above. Complete manufacturer data is missing. In some cases, even the name of the manufacturer, e.g. of the software.
A2: Thanks for the reviewer’s comment. We have revised the manuscript accordingly.
Q3: Line 138: In the sentence beginning "A auto MS/MS was .." should be "An auto MS/MS was ...".
A3: Thanks for the reviewer’s comment. We have revised the manuscript accordingly.
Q4: Line 134: Please state the injection volume and the wavelength in nm at which the GA was detected and quantified.
A4: Thanks for the reviewer’s comment. We have revised the manuscript accordingly. The wavelength was 252 nm, and the injection volume was 20 μL
Q5: Chapter 2.3: Was the calibration curve made for the chromatographic determination of GA content? Please include it in the supplementary material.
A5: Thanks for the reviewer’s comment. Since the current commercial ganoderma acid standard is expensive and incomplete, and the QTOF mass spectrometer used in this study is not suitable for accurate quantitative analysis. We only semi-quantitate and identify the substances in the extract based on the peak area, and we plan to use triple quadrupole tandem mass spectrometry to carry out accurate quantitative detection of various GA components.
References:
- Wu, L,; Liang, W,; Chen, W,; Li, S,; Cui, Y,; Qi, Q,; Zhang, L. Screening and Analysis of the Marker Components in Ganoderma lucidum by HPLC and HPLC-MS(n) with the Aid of Chemometrics. Molecules2017, 22, 584. DOI: 10.3390/molecules22040584
- Cui, M.L,; Yang, H.Y,; He, G.Q. Submerged fermentation production and characterization of intracellular triterpenoids from Ganoderma lucidum using HPLC-ESI-MS. J Zhejiang Univ Sci B2015, 16, 998-1010. DOI: 10.1631/jzus.B1500147
Q6: Line 144: It should be "were kept" and not "were keep".
A6: Thanks for the reviewer’s comment. We have revised the manuscript accordingly.
Q7: Chapter 2.4: Please provide the approval number of your local ethics committee to conduct animal testing.
A7: Thanks for the reviewer’s comment. We have revised the manuscript accordingly.
Institutional Review Board Statement: The animal experimental protocols were conducted in accordance with the guidelines of the Laboratory Animal Welfare and approved by the Committee of Animal Ethics of Institute of Food Science and Technology, Fuzhou University, China (Approval number: FZU-FST-2021-034).
Q8: Lines 158 and 166: Please enter the centrifugation parameters in "g" unit.
A8: Thanks for the reviewer’s comment. We have revised the manuscript accordingly.
Q9: Line 200: Please delete "at" before brackets.
A9: Thanks for the reviewer’s comment. We have revised the manuscript accordingly.
Q10: Lines 253 and 254: Please write the names of the compounds in the middle of the sentence with a lowercase letter (12-hydroxyganoderic acid D, 12-acetoxyganoderic acid F).
A10: Thanks for the reviewer’s comment. We have revised the manuscript accordingly.
Q11: There is something wrong with the page numbering. In the center of the document, the pages are counted from the beginning again.
A11: Thanks for the reviewer’s comment. We have revised the manuscript accordingly.
In short, we have tried our best to improve the manuscript and made some changes in the manuscript. These changes will not influence the content and framework of the paper. We appreciate for Editor/Reviewers’ warm work earnestly, and hope that the corrections would meet with approval.
Once again, thanks for your comments and suggestions.
Sincerely yours,
Xu-Cong Lv and Peng-Hu Liu on behalf of the authors.
Reviewer 2 Report
Title:
The protective effects of ganoderic acids from Ganoderma lucidum fruiting body on alcoholic liver injury and intestinal microbial disorder in mice with excessive alcohol intake
I would like to thank the authors for their effort in investigating the protective effects of ganoderic acids (GAs) from Ganoderma lucidum against liver injury and intestinal microbial disorders in mice with excessive alcohol intake.
Overall, the manuscript is clear and fluent, and I do not see many discrepancies. However, I have detected errors and would recommend revising the English style, and I would also recommend the authors to carefully review the whole manuscript to correct them.
In the following paragraphs, I will provide clear information to improve the manuscript.
Title and abstract
The title is correct and clear. From my point of view, I would recommend the authors to revise the summary. I have found certain errors:
Please eliminate “evidently” may be unnecessary in this sentence. Consider removing.
“a new functional food ingredients” Please consider eliminated the letter “a”. It may not be required with the plural noun “ingredients” in this sentence.
- Introduction
In my opinion, I think the introduction is correct, but I have been able to detect certain errors that I will comment below so that the authors can correct or improve them:
L38: Please change in this sentence “drink that consumed” put “s” in the word drink and the word “that” is not necessary
L45: The verb “increasing” is incorrect form. Please change for “increases”
L51: Please Pluralize the word
L73: “suggest” should be in the past participle form. Consider changing it.
- Materials and methods
In this section I have no consideration. I think this part is well written and I don't see any mistake.
- Results
In general, results are well structured and comprehensive. I will propose some modifications or corrections to the document:
L244: “literatures” is an uncountable noun and should not be made plural. Considering change the noun for “literature, kind of literature, pieces of literature or work of literature”.
L246: Please the word “spectrum” doesn´t seem to fit this context. Consider replacing it with a different one. For example, “spectral”.
L402: The word “injury” may not agree in number with other words in this phrase. Please change for “injuries”.
L405: The word “clearly” is not necessary in this sentence.
This section “3.5 Effects of GA intervention on the composition of intestinal microbiota “discusses that specifically, there was a significant structural change of the intestinal flora along the positive direction of PC2 in the model group. Were there any other changes? In how many days did the changes start to be noticed?
- Discussion
In my view, the discussion is correct but there are some minor problems that follow:
L482: Please change mainly for “is mainly”
L483: Please change “resulted” for “resulting”
Conclusion
The conclusion is well written and clear. But at no time does it talk about future studies or those that are planned for the future. I recommend the authors introduce at least two sentences indicating this.
Tables and Figures
In table 2, the number 17 doesn´t have the same decimals as all other numbers, besides, the first two columns have the title cut off. Please try to get it right.
The figure 5, I recommend the authors, change the images B and D. They are blurred
References
I have no comment on the bibliography
Final Remarks
In general, the article investigates the protective effects and mechanism of ganoderic acids against AML and intestinal microbial disorders induced by excessive alcohol intake. For improvement, the manuscript should be revised according to the above suggestions and those of other reviewers. In my honest opinion, I suggest a minor revision of the article. The authors have done work that provides interesting results.
Author Response
Response to Editor & Reviewers
Journal: Foods
Manuscript ID: foods-1622379
Original title: The protective effects of ganoderic acids from Ganoderma lucidum fruiting body on alcoholic liver injury and intestinal microbial disorder in mice with excessive alcohol intake
Revised title: The protective effects of ganoderic acids from Ganoderma lucidum fruiting body on alcoholic liver injury and intestinal microflora disturbance in mice with excessive alcohol intake
Dear editors and reviewers,
On behalf of my co-authors, I thank you very much for giving us an opportunity to revise the manuscript (Manuscript ID: foods-1622379). We appreciate your positive and constructive comments and suggestions on our manuscript. All of the comments are valuable and helpful for improving our manuscript, as well as the important guiding significance to our future research. Therefore, we have discussed all the comments carefully and made some corrections to meet with approval. The revised content has been marked in red or underlined in the revised manuscript. The main corrections in the manuscript and the response to the editor and reviewers’ comments are as follows:
Editor and Reviewers' comments (Q) and our responses to the comments (A):
Reviewer 2: I would like to thank the authors for their effort in investigating the protective effects of ganoderic acids (GAs) from Ganoderma lucidum against liver injury and intestinal microbial disorders in mice with excessive alcohol intake. Overall, the manuscript is clear and fluent, and I do not see many discrepancies. However, I have detected errors and would recommend revising the English style, and I would also recommend the authors to carefully review the whole manuscript to correct them. In the following paragraphs, I will provide clear information to improve the manuscript.
Q1: Title and abstract: The title is correct and clear. From my point of view, I would recommend the authors to revise the summary. I have found certain errors:
Please eliminate “evidently” may be unnecessary in this sentence. Consider removing.
“a new functional food ingredients” Please consider eliminated the letter “a”. It may not be required with the plural noun “ingredients” in this sentence.
A1: Thanks for the reviewer’s comment. We have revised the manuscript accordingly.
Q2: Introduction: In my opinion, I think the introduction is correct, but I have been able to detect certain errors that I will comment below so that the authors can correct or improve them:
L38: Please change in this sentence “drink that consumed” put “s” in the word drink and the word “that” is not necessary
L45: The verb “increasing” is incorrect form. Please change for “increases”
L51: Please Pluralize the word
L73: “suggest” should be in the past participle form. Consider changing it.
A2: Thanks for the reviewer’s comment. We have revised the manuscript accordingly.
Q3: Materials and methods: In this section I have no consideration. I think this part is well written and I don't see any mistake.
A3: Thanks for the reviewer’s comment.
Q4: Results: In general, results are well structured and comprehensive. I will propose some modifications or corrections to the document:
L244: “literatures” is an uncountable noun and should not be made plural. Considering change the noun for “literature, kind of literature, pieces of literature or work of literature”.
L246: Please the word “spectrum” doesn´t seem to fit this context. Consider replacing it with a different one. For example, “spectral”.
L402: The word “injury” may not agree in number with other words in this phrase. Please change for “injuries”.
L405: The word “clearly” is not necessary in this sentence.
A4: Thanks for the reviewer’s comment. We have revised the manuscript accordingly.
Q5: This section “3.5 Effects of GA intervention on the composition of intestinal microbiota “discusses that specifically, there was a significant structural change of the intestinal flora along the positive direction of PC2 in the model group. Were there any other changes? In how many days did the changes start to be noticed?
A5: Thanks for the reviewer’s comments. We are sorry that our original manuscript was not very clear in this regard. In fact, we did not track the changes in the composition of gut microbiota at different intervention time points, we only focused on the changes in the composition of gut microbiota in mice after 6 weeks of intervention. Our study also referred to other similar studies, which also examined the changes in the composition of gut microbiota in the intervention group compared with the model group and the blank control group after the end of the experimental period.
References:
- Rodriguez-Gonzalez, A,; Vitali, F,; Moya, M,; De Filippo, C,; Passani, M.B,; Orio, L.Effects of Alcohol Binge Drinking and Oleoylethanolamide Pretreatment in the Gut Microbiota. Front Cell Infect Microbiol2021, 11, 731910. DOI: 10.3389/fcimb.2021.731910
- Yang, F,; Wei, J,; Lu, Y,; Sun, Y,; Wang, Q,; Zhang, R.Galacto-oligosaccharides modulate gut microbiota dysbiosis and intestinal permeability in rats with alcohol withdrawal syndrome. Journal of Functional Foods2019, 60, 103423. DOI: https://doi.org/10.1016/j.jff.2019.103423
Q6: Discussion: In my view, the discussion is correct but there are some minor problems that follow:
L482: Please change mainly for “is mainly”
L483: Please change “resulted” for “resulting”
A6: Thanks for the reviewer’s comment. We have revised the manuscript accordingly.
Q7: Conclusion: The conclusion is well written and clear. But at no time does it talk about future studies or those that are planned for the future. I recommend the authors introduce at least two sentences indicating this.
A7: Thanks for the reviewer’s comment. We have revised the manuscript accordingly.
“This study reveals that GA has potential profitable effects in taking precautions against alcohol-induced liver injury, and is estimated to be a promising functional food ingredient. In further studies, we need to illustrate the protective mechanism of GA intervention on ALD through multiomics technology.”
Q8: Tables and Figures: In table 2, the number 17 doesn´t have the same decimals as all other numbers, besides, the first two columns have the title cut off. Please try to get it right. The figure 5, I recommend the authors, change the images B and D. They are blurred
A8: Thanks for the reviewer’s comment. We have revised the manuscript accordingly.
Q9: References: I have no comment on the bibliography
A9: Thanks for the reviewer’s comment.
Q10: Final Remarks: In general, the article investigates the protective effects and mechanism of ganoderic acids against AML and intestinal microbial disorders induced by excessive alcohol intake. For improvement, the manuscript should be revised according to the above suggestions and those of other reviewers. In my honest opinion, I suggest a minor revision of the article. The authors have done work that provides interesting results.
A10: Thanks for the reviewer’s comment. We have revised the manuscript accordingly.
In short, we have tried our best to improve the manuscript and made some changes in the manuscript. These changes will not influence the content and framework of the paper. We appreciate for Editor/Reviewers’ warm work earnestly, and hope that the corrections would meet with approval.
Once again, thanks for your comments and suggestions.
Sincerely yours,
Xu-Cong Lv and Peng-Hu Liu on behalf of the authors.